# Vocabulary-informed Visual Feature Augmentation for One-shot Learning

## Abstract

A natural solution for one-shot learning is to augment training data to handle the data deficiency problem. However, directly augmenting in the image domain may not necessarily generate training data that sufficiently explore the intra-class space for one-shot classification. Inspired by the recent vocabulary-informed learning, we propose to generate synthetic training data with the guide of the semantic word space. Essentially, we train an auto-encoder as a bridge to enable the transformation between the image feature space and the semantic space. Besides directly augmenting image features, we transform the image features to semantic space using the encoder and perform the data augmentation. The decoder then synthesizes the image features for the augmented instances from the semantic space. Experiments on three datasets show that our data augmentation method effectively improves the performance of one-shot classification. Extensive study shows that data augmented from semantic space are complementary with those from the image space, and thus boost the classification accuracy dramatically. Source code and dataset will be available.

## 1 Motivation and Introduction

The success of recent machine learning (especially the deep learning) greatly relies on the training process on hundreds or thousands of labelled training instances of each class. However in practice, it might be extremely expensive or infeasible to obtain many labelled data, e.g. for objects in dangerous environment with limited access. On the other hand, human can recognize an object category easily with only a few shots of training examples Thrun (1996). Inspired by such an ability of humans, one-shot learning aims at building classifiers from a few or even a single example.

The major obstacle of learning good classifiers in one-shot learning setting is the lack of enough training data. Thus a natural recipe for one-shot learning is to augment the data, which has been conducted in various ways. The dominant approach adopted by previous work is to bring in more images Krizhevsky et al. (2012) for each category as training data. These additional augmented training images could be borrowed from unlabelled data Fu et al. (2015) or other relevant categories Wang & Hebert (2016a;b); Li & Hoiem (2016); Lim et al. (2011) in an unsupervised or semi-supervised fashion; however the semantic signals of augmented data are often noisy and unreliable and may suffer from *negative transfer* when the augmented data are from different classes. On the other hand, synthetic images rendered from virtual examples Movshovitz-Attias (2015); Park & Ramanan (2015); Movshovitz-Attias et al. (2015); Dosovitskiy et al. (2015); Zhu et al. (2016b); Opelt et al. (2006) are semantically correct but require careful domain adaptation to transfer the knowledge to the real image domain. In contrast, we propose to directly augment training data in the image feature domain rather than the original image. Augmenting data in image feature domain allows us to interact with useful discriminative signals more directly. The most similar work to us is Zhu et al. (2016b); Opelt et al. (2006), where the feature patches (*e.g.* HOG) of the object parts are combined to synthesize new feature representation. However, their approach requires strong heuristics and spatial information to learn the combination. On the contrary, we augment data in compact deep learning based feature space, which is stronger for classification but contains limited the spatial information.

A straightforward approach to augment image feature is to add random vectors to the feature of each single training image. However, the cutting plane for the classification is usually not in a regular shape, *e.g.* a hyper ball, and such a simple disturbance, *e.g.* sampled from Gaussian distribution, may not sufficiently explore the intra-class space for each individual category. Our idea of data

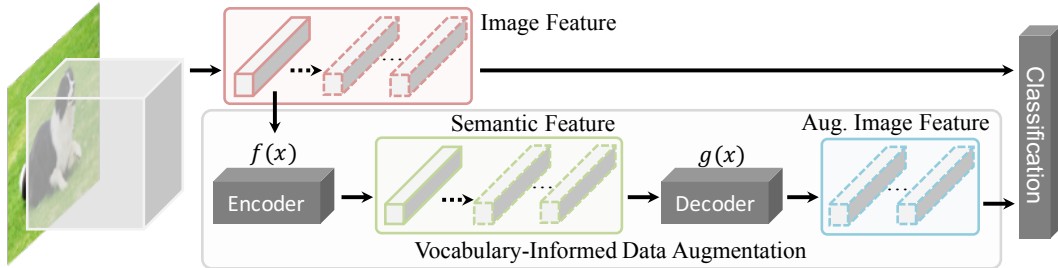

Figure 1: **Overview of our framework.** We extract image feature and project it to the semantic space using the encoder. After augmenting data in semantic space, we use the decoder to obtain the corresponding image feature for augmented semantic feature. Both real and augmented data are used to train the classification model.

augmentation is to bring in the additional knowledge by leveraging the semantic space pre-trained by linguistic models such as google word2vec Mikolov et al. (2013) from a huge repository of linguistic corpora. In semantic space, similar concepts are supposed to have similar feature, such that the overall space demonstrates superior manifold continuity over semantic meaning, which is ideal for data augmentation. To leverage such good property, we learn a mapping ($f(\mathbf{x})$) to transform the image feature into the semantic space, where we add Gaussian random vectors on the semantic feature from the only training image. We then learn an inverse mapping ($g(\mathbf{x})$), from the semantic to image, to convert the synthesized semantic features into image features, which are then fed into the classification model during the training. In experiment, we found the data augmented in semantic space complements the data augmented in the image feature domain. Each set of augmented data can improve the performance individually, and the best performance is achieved by adding them together.

The semantic space also allows us to leverage the knowledge resides in the vocabulary space. Like in the image feature domain where not all the spots corresponds to a meaningful input image Sabour et al. (2016), it might also be the case for the semantic space where not all the points represents a real concept. As such, we adopt a non-parametric approach to synthesize training data in the semantic domain leveraging the huge repository of linguistic corpora. Specifically, we calculate the semantic feature for a large number of vocabulary and phrases Mikolov et al. (2013) as a pool of candidate semantic features, from where we search for nearest neighbors for the given training instance as the augmented data. The semantic feature augmented in this way is then guaranteed to represent certain concepts that can be described by some words. Such nearest neighbors set is semantically similar to the training data and can thus help increase the intra-class coverage and differentiate the one-shot categories. For example, as visualized in Fig. 5(b), the neighbors of training instances in "killer whale" are corresponding to the semantic features of related topics, such as "sea lions" and "orcas", which are different from the other one-shot training categories such as "grizzly bear".

The mapping between feature spaces, $f(\mathbf{x})$ and $g(\mathbf{x})$, can be learned with a variety of models. In practice, we found that the auto-encoder is an elegant and effective framework to learn two mapping at the same time. Taken an image feature as input, the encoder ($f(\mathbf{x})$) learns to produce semantic feature, and the decoder ($g(\mathbf{x})$) learns to recover the image feature. The auto-encoder also allows an end-to-end training, which optimizes the image feature extractor, feature space mappings, and the classification model jointly.

The contribution of this paper are mainly in three aspects. First, we propose to perform data augmentation directly in compact and discriminative image feature space rather than the original image domain. Second, we leverage the semantic space for data augmentation, which brings additional chance to explore the intra-class space thoroughly. Last, we propose a simply yet elegant deep learning architecture that allows end-to-end training for optimal performance.

## 1.1 RELATED WORK

One-shot learning is inspired by the celebrated aspects of human learning Jankowski et al. (2011); Lake & Salakhutdinov (2013) of being able to learn about new concepts from very few examples. Generalizing to recognize new classes with only few examples Bart & Ullman (2005) is quite beyond the capability of typical machine learning tools, which nevertheless require hundreds or thousands of training examples. The gradient-based strategies inevitably re-learn the parameters from a few scraps

of information and leads to catastrophic fails of generalizability of models. We can roughly divide the existing one-shot learning into two groups as discussed below in term of whether transferring extra knowledge to help learning one-shot tasks.

### 1.1.1 Direct One-shot Learning Approaches

One-shot learning in this category can be implemented via instance-based learning (such as nearest neighbour or K-nearest neighbour), non-parameteric methods Fei-Fei et al. (2003; 2006); Tommasi & Caputo (2009), or deep generative models Rezende et al. (2016); Santoro et al. (2016), or Bayesian auto-encoders Kingma & Welling (2014). These approaches directly learn the one-shot learning model, rather than assume an auxiliary set of helping transfer knowledge as in Sec. 1.1.2. Thus these works only employ a rich class of generative models to explain the observed data, rather than directly augmenting the training data in the feature space like our work.

### 1.1.2 Knowledge transfer based One-shot Learning Approaches

This category of one-shot learning can also be explored via learning to learn Thrun (1996), which is also known as transfer learning Pan & Yang (2010), or meta-learning JVilalta & Drissi (2002). The key insight is to transfer the knowledge from auxiliary data/categories to recognize new categories with few examples by either sharing features Bart & Ullman (2005); Hertz et al. (2016); Fleuret & Blanchard (2005); Amit et al. (2007); Wolf & Martin (2005); Torralba et al. (2007), or semantic attributes Fu et al. (2013); Lampert et al. (2013); Rohrbach et al. (2013; 2010), or contextual information Torralba et al. (2010). Especially, the recent deep meta-learning has received increasing attention for one-shot learning Santoro et al. (2016); Bertinetto et al. (2016); Koch et al. (2015); Habibian et al. (2014a;b); Zheng et al. (2016); Vinyals et al. (2016); Zhang et al. (2016); Lake & Salakhutdinov (2013).

One general strategy of such category is to learning an embedding space by different tools such as neural networks, *e.g.*, siamese network Bromley et al. (1993); Koch et al. (2015) or discriminative methods (*e.g.*, Support Vector Regressors (SVR) Farhadi et al. (2009); Lampert et al. (2013); Kienzle & Chellapilla (2006)), metric learning methods Quattoni et al. (2008); Fink (2005); Vinyals et al. (2016), or kernel embedding method Wolf et al. (2009); Hertz et al. (2016). Particularly, one of most common embedding ways is semantic embedding which is normally explored by projecting the visual features and semantic entities into a common *new* space. Such projections can use various forms of corresponding loss functions, such as ReViSE Tsai et al. (2017), SJE Akata et al. (2015), WSABIE Weston et al. (2011), ALE Akata et al. (2013), DeViSE Frome et al. (2013), and CCA Fu et al. (2014).

Recently, transfer based one-shot learning approaches have been quite heavily studied in machine learning community Vinyals et al. (2016); Snell et al. (2017); Santoro et al. (2016); Bertinetto et al. (2016). However, this type of approaches still requires and relies on the well-labelled, well-organized and large enough auxiliary set of similar visual information, while such auxiliary set requires expertise knowledge and might be expensive to obtain in many real world scenario. In contrast, the idea of our framework solves one-shot learning from the perspective of directly augmenting visual features and thus can also work when we do not have enough auxiliary dataset.

### 1.1.3 One-shot learning by augmenting training instances

In principle, augmenting training instances have been exploited in supervised learning settings by many previous works Krizhevsky et al. (2012); thus training data augmentation can also be employed to alleviate the problem of lacking instances in one-shot learning settings. In particular, this type of approaches have been explored via various ways: (1) Learning one-shot models by utilizing the manifold information of large amount of unlabelled data in a semi-supervised or transductive learning way Fu et al. (2015); (2) Adaptively learning the one-shot classifiers from off-shelf trained models Wang & Hebert (2016a;b); Li & Hoiem (2016); (3) Borrowing examples from relevant categories Lim et al. (2011) or semantic vocabularies Fu & Sigal (2016); Ba et al. (2015) to augment the training set; (4) synthesizing new labelled training data by rendering virtual examples Movshovitz-Attias (2015); Park & Ramanan (2015); Movshovitz-Attias et al. (2015); Dosovitskiy et al. (2015) or composing synthesized representations Zhu et al. (2016b); Opelt et al. (2006) or distorting existing training examples Krizhevsky et al. (2012); (5) Generating new examples by Generative Adversarial Networks

(GANs) Zhu et al. (2017; 2016a); Goodfellow et al. (2014); Reed et al. (2016); Radford et al. (2016); Mao et al. (2017); Durugkar et al. (2017); Huang et al. (2017)

Previous methods may suffer from different problems: the manifold information may be not useful for one-shot learners; there may be *negative transfer* when the off-shell models or relevant categories are very different from one-shot classes; rendering, composing new virtual examples or distorting training examples may either need experience of domain expertise or only works for different domains. GANs based approaches mostly focused on learning good generators to synthesize "realistic" images to "cheat" the discriminators, while the synthesized images are not learned to preserve the discriminative information, which, is in contrast to our network structure, where the discriminative instances are synthesized in visual feature domain.

## 2    METHODOLOGY

In this section, we introduce the detail of our algorithm for one-shot classification. In a high level, our approach relies on continuity of the manifold in semantic space to synthesize training data, which is complementary with the data augmented directly from the image feature domain, and thus increase the generalization capability of the model. To achieve this, we learn a two-sided mapping between the image feature space and the semantic space in a single deep auto-encoder. With the help of these mappings, the data augmentation can be done in the semantic space, which is combined with the data augmented in the image space together to train the one-shot classification model.

**Problem setup.** The training image dataset $D_s = \{\mathbf{I}_i, z_i\}_{i=1}^{N_s}$ of $N_s$ samples. $\mathbf{I}_i$ indicates the raw pixel matrix of image $i$. $z_i \in \mathcal{W}_s$ is a class label and the corresponding vocabulary $w_i$ taken from the vocabulary set $\mathcal{W}$. The vocabulary $\mathcal{W}$ is learned by word2vec Mikolov et al. (2013) on large-scale corpus; each vocabulary entity $w \in \mathcal{W}$ is projected as a semantic vector $\mathbf{u} \in \mathbb{R}^d$.

### 2.1    DEEP AUTO-ENCODER EMBEDDING NETWORK

The deep auto-encoder embedding network learns the mapping between the Image Feature Space (IFS) to the Semantic Space (S-S). Linear mapping has been used in previous work Fu & Sigal (2016). However, practically, such a mapping should be highly non-linear; and thus a deep auto-encoder framework with supervision on the bottleneck layer is introduced here. The whole deep auto-encoder includes two main components, i.e. *image feature extractor*, and *auto-encoder.*

**Image feature extractor** converts the raw images into image feature vectors by the pre-trained deep convolutional network. We use pre-trained VGG-16 network Chatfield et al. (2014) which consists of 13 convolutional and 3 fully connected layers. For an input image $\mathbf{I}_i$, the image feature extractor output the 4096-dimensional feature vector $\mathbf{x}_i$ from the fc7 layer.

**Auto-encoder network** can be divided into the encoder and decoder. The encoder is composed of five fully connected layers with the 4096, 2048, 1024, 512, and 256 neurons accordingly followed by non-linear kernel RELU. The structure is detailed in Fig. 2(a). The encoder part learns the mapping from IFS to S-S, *i.e.*, $\hat{\mathbf{u}}_i = f(\mathbf{x}_i)$ by projecting the image feature vectors to corresponding semantic feature vectors. The decoder has exactly the symmetry architecture with the encoder learning the map from S-S to IFS, *i.e.*, $\hat{\mathbf{x}}_i = g(\mathbf{u}_i)$. It is attached after the semantic feature to reconstruct the image feature. Note that different from the standard unsupervised auto-encoder, our deep auto-encoder network tries to embed the IFS and S-S. In other word, the latent representation space learned by auto-encoder network is the semantic space, which should be posed as additional supervised information when learning such an embedding. Thus the loss function is

$$J(\Theta) = \mathbb{E}_{\mathbf{x}_i \in D_s} \left[ (\mathbf{x}_i - \hat{\mathbf{x}_i})^2 + (\hat{\mathbf{u}}_i - \mathbf{u_i})^2 \right] + \lambda P(\Theta) \tag{1}$$

where $\Theta$ indicates the parameter set of auto-encoder network and $P(\cdot)$ is the regularized penalty function. We use $L_2$−penalty in experiments. The Image feature extractor and auto-encoder networks are trained on $D_s$.

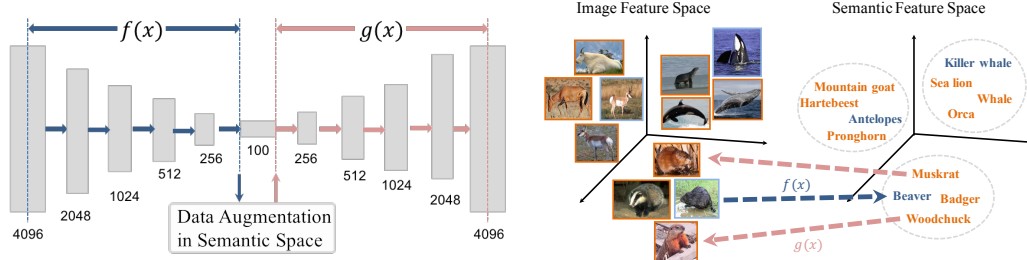

(a) Auto-encoder for feature mapping      (b) Data Augmentation in Semantic Space

Figure 2: **(a) Auto-encoder Network.** We use fully connected layers and the number indicates the amount of neurons. Each fully connected layer is followed by non-linear kernel RELU. **(b) Augmenting data in semantic space using auto-encoder.** The training image (in blue box) feature is mapped to the semantic space (blue text), where nearest neighbor (red text) are searched. The neighboring semantic features are then projected back to the image feature domain as the augmented features (red box). Noted that for purpose of visualization, we demonstrate with images and words which are actually feature vectors.

## 2.2 DATA AUGMENTATION

We perform data augmentation in two domains and combine them together to train the classification model. Not necessary to have the augmented images, our goal is to generate more image features as the training data for each category.

**Augmentation via Image Feature Space.** A natural way of augmenting the training data is via the image feature space. Given the features $\mathbf{x}_i$ of training instance $i$, the augmented data $\mathbf{x}^F$ can be sampled from the corresponding *Feature Gaussian*, i.e., $\mathbf{x}_i^F \sim \mathcal{N}\left(\mathbf{x}_i, \sigma^F \mathbf{E}\right)$, $\sigma^F \in \mathbb{R}$ is the variance of each feature dimension; $\mathbf{E}$ is the identity matrix. $\sigma^F$ controls the deviation of noise added. To make the augmented feature vector $\mathbf{x}_i^F$ still be representative enough of the class of $\mathbf{x}_i$, we empirically set $\sigma^F$ as $10\%$ of the distance between $\mathbf{x}_i$ and its nearest training instance $\mathbf{x}_j$ ($z_i \neq z_j$) as this gives the best performance.

**Augmentation via Semantic Space.** We introduce two types of augmentation in semantic space.

(1) Semantic Gaussian: the same as Feature Gaussian, we can explore the continuity of the manifold in the semantic space. Given the features $\mathbf{x}_i$ of training instance $i$, the augmented data $\mathbf{x}^G$ can be sampled from the corresponding *Semantic Gaussian*, i.e., $\mathbf{x}_i^G \sim g\left(\mathcal{N}\left(\hat{\mathbf{u}}_i, \sigma^G \mathbf{E}\right)\right)$, $\sigma^G \in \mathbb{R}$ is the variance of each semantic dimension. The $\sigma^G$ is also empirically set as the $10\%$ of the distance between $\hat{\mathbf{u}}_i$ and its nearest semantic vector $\hat{\mathbf{u}}_j$ ($z_i \neq z_j$).

(2) Semantic Neighborhood (Fig 2(b)): Note that the semantic space is not uniformly dense; the distributions of words of vocabulary may depend on the frequency of words in the linguistic corpora. This indicates that the intra-class deviation of different category might be different, and a gaussian distribution with a fixed standard deviation may not suit all the categories. Thus, given the feature $\mathbf{x}_i$ of training instance $i$, the augmented data $\mathbf{x}^H$ can be sampled from the corresponding *Semantic Neighborhood* , i.e., $\mathbf{x}_i^H \sim g\left(\mathbf{u}_j\right)$, $\mathbf{u}_j \in Neigh\left(\hat{\mathbf{u}}_i\right)$, $w_j \in \mathcal{W}$ and $Neigh\left(\hat{\mathbf{u}}_i\right) \subseteq \mathcal{W}$. The $Neigh\left(\hat{\mathbf{u}}_i\right)$ indicates the nearest neighborhood vocabulary set of $\hat{\mathbf{u}}_i$. These neighbors correspond to the most similar examples in the semantic space to our real training instance, and thus can be used to augment training data.

## 2.3 ONE-SHOT CLASSIFICATION AND TRAINING STRATEGY

As mentioned above, the image feature extractor and auto-encoder networks are trained from $D_s$; and training set together with augmented dataset are employed to train the one-shot classifiers. Specifically, we can now employ the augmented dataset $\left\{\left[\mathbf{x}_i; \mathbf{x}_i^A\right], z_i\right\}$, $\mathbf{x}_i^A \in \left\{\mathbf{x}_i^F, \mathbf{x}_i^G, \mathbf{x}_i^H\right\}$ to better train the one-shot classification model. Specifically, the original and augmented image features are grouped together and fed into the classification network. As shown in Fig. 1, the one-shot classification model consists of two fully connected layer (with ReLu) and a soft-max layer for classification.

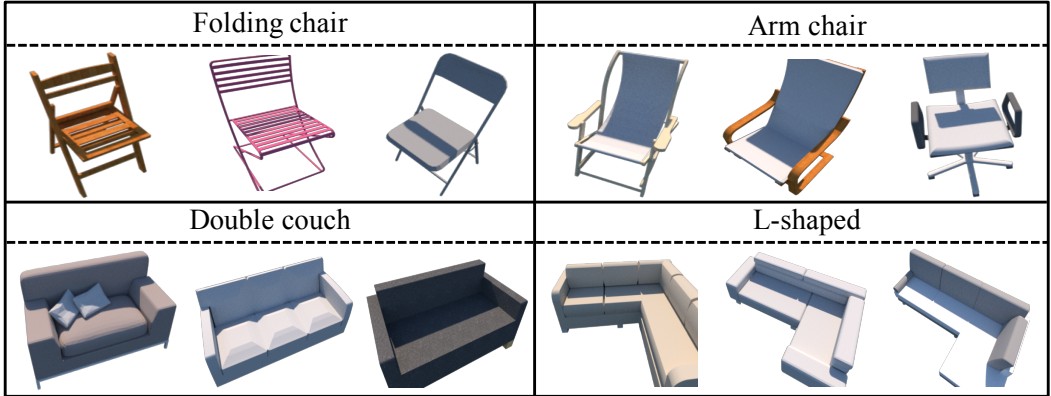

Figure 3: **Example images from ShapeNet Multiview dataset.** As a reflection of the real indoor scene, the categories are chosen to cover various types of furniture, whereas including fine-grained categories as well.

## 3 EXPERIMENTS

In this section, we evaluate our method on three datasets which cover a variety of cases in real application. We also conduct analysis and ablation study to demonstrate the effectiveness of our data augmentation method.

### 3.1 EXPERIMENTAL SETUP

**Dataset.** We choose three datasets that are representative for different types of classification problem. To highlight the effectiveness of our data augmentation, we focus on more challenging scenario where the difference between classes are comparatively smaller than usual.

*Animals with Attributes (AwA) dataset*: AwA consists of 50 classes of animals (30,475 images in total). As the classes are all about animal, the difference between classes are smaller than arbitrary cases, such as table and dog in which data augmentation would be comparatively easy.

*Cub-bird 200-2011 dataset* Wah et al. (2011) contains 11,788 images of 200 categories of birds. This dataset is especially designed for fine-grained recognition, where the data augmentation is extremely hard as the difference between categories are fairly small.

*ShapeNet Multiview dataset*: Indoor scene understanding is yet another scenario where one-shot learning is especially useful. The capability to recognize furniture in a few shots is a desirable feature for indoor robots. To evaluate in this scenario, we create a synthetic dataset containing 4660 photo-realistic rendering of indoor furnitures. We select 233 object 3D models from 24 fine-grained furniture related classes from ShapeNet Core V2. For each model, we set up environment illumination and render photo-realistic color images from 20 cameras looking at the object from different direction and distance. Fig. 3 shows some example images from our synthetic indoor furniture dataset.

**Settings and Parameters.** The dropout rate and learning rate of the auto-encoder network is set to 0.8 and $1e^{-4}$ respectively to prevent overfitting. The batch size is set to 50. The network is trained by Stochastic Gradient Descendent (SGD), and usually converges in 300 epochs. The learning rate of image feature extractor is set to $1e^{-8}$ in the fine-tuning step. To prevent randomness due to the small training set, all experiments are repeated for 10 times, and the average performances over all runs are reported. We employ the mean accuracy (i.e. mean of the diagonal of the confusion matrix) to evaluate the classification. The source codes and the synthetic dataset will be released. The same 100-dimensional vocabulary dictionary used in Fu & Sigal (2016) is used here.

### 3.2 DOES THE DATA AUGMENTATION MATTER?

To answer this question, we conduct the experiments in one-shot learning scenario on all three datasets.

| (%) | No data augmentation | Number of instances augmented | | | Chance-level |
|---|---|---|---|---|---|
| | | 1 | 5 | 10 | |
| AwA | 17.50 | 27.98 | 30.43 | **30.57** | 2.50 |
| Cub-bird | 5.29 | 7.16 | 7.89 | **8.21** | 0.50 |
| Shapenet | 14.21 | 17.04 | **18.51** | 18.18 | 4.17 |

Table 1: **The classification accuracy of one-shot learning on three dataset.** "Number of instances augmented" indicates how many virtual instances are generated from one training instance in each type of data augmentation method. The standard deviation is $0.5 \sim 1.3\%$.

**Comparing with/without data augmentation.** For each dataset, we randomly sample 1 instance from each category for training, and take all the other images for testing. All three types of data augmentation are used, and the results are shown in Tab. 1. We vary the number of instances augmented for each training instance. As a standard baseline for comparison, we also show the performance with the same classification network introduced in Sec. 2.3 without any data augmentation.

Most importantly, on all of the three datasets, training with our augmented data significantly improves the classification accuracy over the cases without any data augmentation. Particularly on AwA dataset, our framework dramatically increase the accuracy by 13.07 absolute point, which almost double the baseline. This validates that our deep auto-encoder embedding framework effectively augments the training features which bring in additional information that can be leveraged by the classification model. Note that all the results in Tab. 1 are significantly better than the accuracy of chance-level.

Fig. 5(a) shows the confusion matrix on AwA before (top) and after (bottom) augmenting 10 instances per class. Without data augmentation, the confusion matrix has vertical bands indicating consistent misclassification; in contrast, the one with data augmentation has the clearer diagonal structure, which demonstrates the data augmentation helps to resolve the ambiguity across categories.

### 3.3 DOES THE SEMANTIC SPACE AUGMENTATION MATTER?

To answer this question, we further conduct the one-shot learning experiment to evaluate the effectiveness of each type of data augmentation. Specifically, on each dataset, we use 1 instance per class for training, and each training instance is augmented by 5 instances. We compare the results of (1) without data augmentation (No Aug); and data augmentation by (2) Feature Gaussian (FeatG), (3) Semantic Gaussian (SemG), (4) both Feature Gaussian and Semantic Gaussian (FeatG+SemG), (5) Semantic Neighborhood (SemN); (6) Feature Gaussian and Semantic Neighborhood (FeatG+SemN); and (7) all together (FeatG+SemN+SemG). The results are shown in Tab. 2.

We notice that on all three datasets, the results of combining all three types of augmentation achieve the best performance, outperforming all the other methods by a clear margin. Specifically, the result of each single type of data augmentation can already beat the situation with no data augmentation, and combining data augmented from both image feature space and semantic space outperforms the case of using only one feature space. This indicates that these three types of data augmentation are complementary to each other. The combinations of Feature Gaussian and Semantic Neighborhood (FeatG+SemN) have the second best results on all three dataset. This shows that these two types of data augmentation are most complementary to each other.

Fig. 5(b) shows the visualization of semantic space. Intuitively, the neighborhood vocabulary of each training prototype can help differentiate the corresponding prototypes with the other training prototypes. Thus these vocabulary in the neighborhood set of one prototype, *i.e.* Semantic Neighborhood, can be used to reliably augment the data.

To visually demonstrate the effect of exploring semantic space, we visualize some classification results on AWA in Fig. 4. Each training image is followed by three testing images that are misclassified as the categories below before the augmentation, which are later correctly classified after the augmentation. These testing images are visually similar to other categories, but different with the only training image on the left, and thus are extremely hard to recognize correctly. Our augmentation explore the semantic space and build connection between the testing images and the training image.

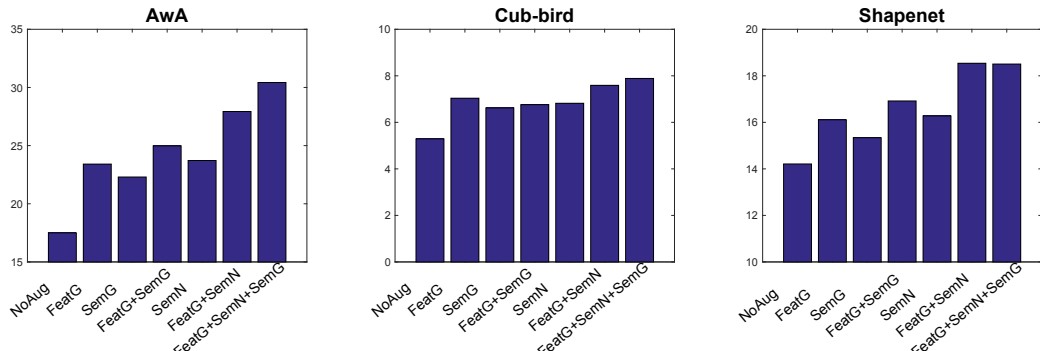

Table 2: **The results of combining different types of data augmentation in our framework.** "FeatG", "SemG", and "SemN" denote the data augmentation methods by Feature Gaussian, Semantic Gaussian and Semantic Neighborhood respectively. Combining augmented data from both space achieves the best performance.

| Number of training instances/class | Models | No data augmentation | Number of instances augmented | | |
|---|---|---|---|---|---|
| | | | 1 | 5 | 10 |
| 1 | NN | $17.50 \pm 2.8$ | $27.98 \pm 1.3$ | $30.43 \pm 0.9$ | $\mathbf{30.57} \pm 0.8$ |
| | LR | $5.07 \pm 1.1$ | $6.19 \pm 1.5$ | $7.42 \pm 0.2$ | $\mathbf{8.31} \pm 0.4$ |
| | Near-N | $21.96 \pm 1.2$ | $26.45 \pm 0.9$ | $\mathbf{27.26} \pm 1.6$ | $26.85 \pm 0.4$ |
| | SVR | $18.28 \pm 1.6$ | $24.78 \pm 1.4$ | $\mathbf{24.96} \pm 0.8$ | $24.83 \pm 0.1$ |

Table 3: **One-shot Learning Performance on AwA dataset w.r.t number of training instances per category.** NN, LR, Near-N and SVR indicate neural network, logistic regression, nearest neighbour and support vector regressor for classification and feature mapping. The chance rate is 2.5.

| Number of training instances/class | Models | No data augmentation | Number of instances augmented | | |
|---|---|---|---|---|---|
| | | | 1 | 5 | 10 |
| 3 | NN | 35.87 | 45.69 | 45.61 | **47.15** |
| | LR | 16.77 | **19.06** | 18.23 | 18.65 |
| | SVR | 25.13 | **32.92** | 32.63 | 30.41 |
| 5 | NN | 47.11 | **54.75** | 53.78 | 51.36 |
| | LR | 25.92 | **26.72** | 26.24 | 26.70 |
| | SVR | 37.73 | 37.87 | 36.92 | **40.10** |

Table 4: **Few-shot learning performance on AwA dataset w.r.t number of training instances per category.** NN, LR, and SVR indicate neural network, logistic regression, and support vector regressor for classification and feature mapping. The standard deviation is $0.5 \sim 1.3\%$.

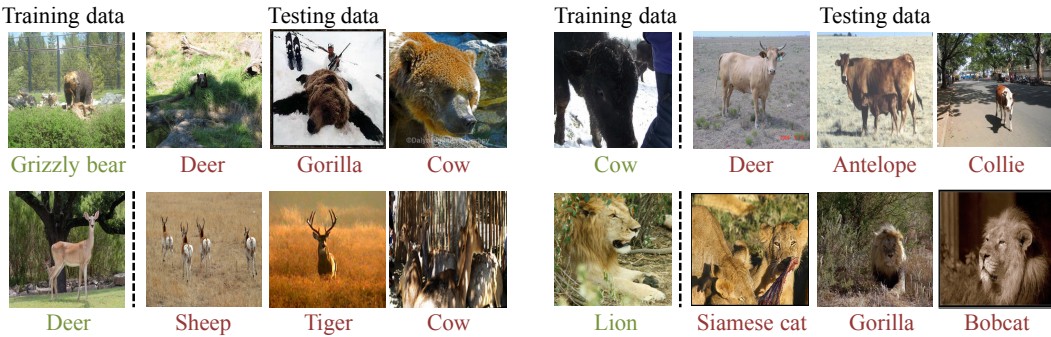

Figure 4: **Visualization of the classification results.** Each training image is followed by three testing images that are misclassified as the categories below before the augmentation, which are later correctly classified after the augmeantation. These testing images are visually different with the only training data, ambiguous with other categories, and thus hard to recognize. Our augmentation can help to correctly recognize them.

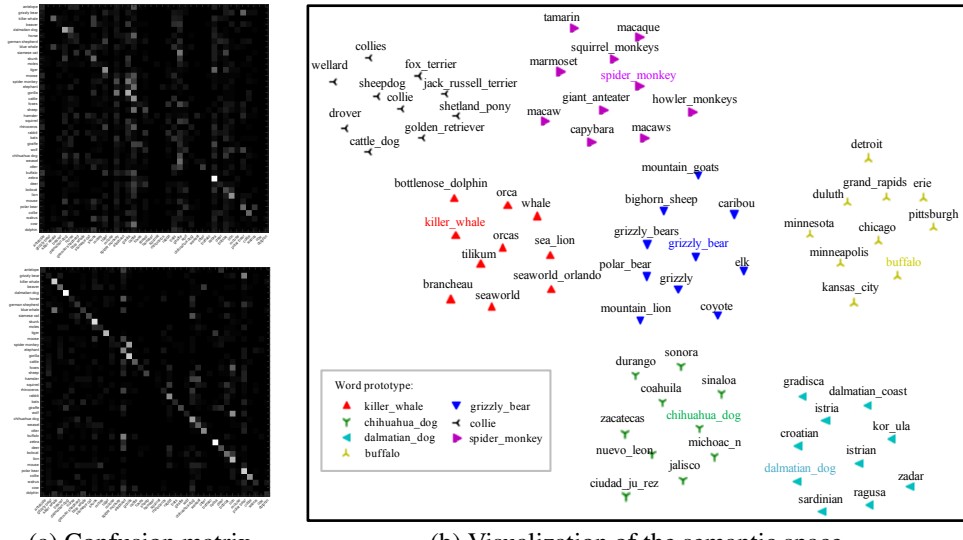

(a) Confusion matrix       (b) Visualization of the semantic space

Figure 5: **(a) Confusion matrix on AWA before (top) and after (bottom) data augmentation.** Learning with our augmented data shows clear diagnal pattern in the confusion matrix. **Visualization of the semantic space.** We show the t-SNE visualization of the semantic feature space. The words in boxes are the mapping of training image in the semantic space, and close neighbors are shown. The neighborhoods extend the single training data to a space semantically meaningful.

### 3.4 DOES DATA AUGMENTATION STILL MATTER WHEN MORE TRAINING DATA IS AVAILABLE?

We evaluate on AWA to demonstrate the behavior of our data augmentation approach with different number of training instances per category. We compare the one-shot learning results in Tab. 3 and few-shot learning results in Tab. 4. For each training instance, we augment 1, 5, 10 instances respectively. Surprisingly, our data augmentation managed to improve the performance dramatically for all the cases, though the performance get saturated earlier with more training instances given for each category, *i.e.* augmenting 1 instance per category gives the best performance for 5 instance per class. This validates that our data augmentation can effectively utilize the information from each individual training instance to generate good augmented data. Interestingly, the performance does not keep increasing all the time as more instances are augmented, as shown in Tab. 4. It is largely due to the increasing possibility of augmenting non-discriminative noisy instances with a large number of instances augmented.

### 3.5 DOES THE DEEP LEARNING ARCHITECTURE MATTER?

To demonstrate if our deep learning architecture is optimal and suitable for data augmentation, we run the same experiment in the previous section but replacing deep auto-encoder embedding with other supervised model, such as logistic regression (LR)[1], and support vector regressor (SVR), and the results are shown in Tab. 3 as well.

As can be seen, our data augmentation method consistently improve the classification accuracy with either LR or SVR. This demonstrates that our method is not constrained to work only with deep learning architecture, but can be generalized to work ubiquitously well with many different kinds of models. Nevertheless, the deep auto-encoder based model still shows superior base performance with no data augmentation and brings in much larger performance gain compared to the other two traditional models. This implies that the auto-encoder achieves a better mapping between two feature spaces, and the end-to-end learning allows joint optimization of the whole system in order to achieve optimal performance.

---

[1]Note SVM is not used here since it is unstable when trained by 1 training instance per class.

### 3.6 Does the data augmentation can also help the other state-of-the-art one-shot learning methods?

Note that our vocabulary-informed feature augmentation framework is actually orthogonal to the one-shot learning approaches. Thus it is also interesting to see whether our augmented features can also be used to help those previous work. To demonstrate this point, we run the codes of SS-Voc Fu & Sigal (2016) on the 10-way target classes from AWA dataset without and with the data augmentation. The classification accuracy is improved from 0.432 to 0.453 after using the data agumentation. This validates that the our feature augmentation complements typical one-shot learning algorthms.

## 4 Conclusion

We propose a vocabulary-informed data augmentation method for one-shot classification. We demonstrate that the data augmented from the semantic space complements the image feature domain and thus can further increase the overall classification performance. Future research can be held by investigating more generalized feature representation, extending the current system to zero-shot and open-set learning problem, or deploying in real applications.

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
