# OpenReview forum: "VOCABULARY-INFORMED VISUAL FEATURE AUGMENTATION FOR ONE-SHOT LEARNING"
_ICLR.cc/2018/Conference — Reject_

### Official Review · AnonReviewer3 · 2017-11-28
**Method needs to be clarified; experiments need to be improved**

**Rating:** 4
**Confidence:** 3

**Review:**

This paper proposes a feature augmentation method for one-shot learning.  The proposed approach is very interesting. However, the method needs to be further clarified and the experiments need to be improved.

Details:
1. The citation format used in the paper is not appropriate, which makes the paper, especially the related work section, very inconvenient to read.

2. The approach:
(1) Based on the discussion in the related work section and the approach section, it seems the proposed approach proposes to augment each instance in the visual feature space by adding more features, as shown by [x_i; x_i^A] in 2.3.  However, under one-shot learning, won’t this  make each class still have only one instance for training?

(2) Moreover, the augmenting features x_i^A (regardless A=F, G, or H), are in the same space as the original features x_i. Hence x_i^A is rather an augmenting instance than additional features. What makes feature augmentation better than instance augmentation?

(3) It is not clear how will the vocabulary-information be exploited? In particular, how to ensure the semantic space u to be same as the vocabulary semantic space? How to generate the neighborhood in Neigh(\hat{u}_i) on page 5?

3.  In the experiments:
(1) The authors didn’t compare the proposed method with existing state-of-the-art one-shot learning approaches, which makes the results not very convincing.

(2) The results are reported for different numbers of augmented instances. Clarification is needed.

---

### Official Review · AnonReviewer2 · 2017-11-28
**This paper proposes a semantic approach for data augmentation, but the experiments are weak/unconvincing at this state**

**Rating:** 6
**Confidence:** 4

**Review:**

This paper proposes a (new) semantic way for data augmentation problem, specifically targeted for one-shot learning setting, i.e. synthesizing training samples based on semantic similarity with a given sample . Specifically, the authors propose to learn an autoencoder model, where the encoder translates image data into the lower dimensional subspace of semantic representation (word-to-vec representation of image classes), and the decoder translates semantic representation back to the original input space. For one-shot learning, in addition to a given input image, the following data augmentation is proposed: a) perturbed input image (Gaussian noise added to input image features); b) perturbed decoded image; c) perturbed decoded neighbour image, where neighbourhood is searched in the semantic space.
The idea is nice and simple, however the current framework has several weaknesses:
1. The whole pipeline has three (neural network) components: a) input image features are extracted from VGG net pre-trained on auxiliary data; 2) auto-encoder that is trained on data for one-shot learning; 3) final classifier for one-shot learning is learned on augmented image space with two (if I am not mistaken) fully connected layers. This three networks need to be clearly described; ideally combined into one end-to-end training pipeline.
2. The empirical performance is very poor. If you look into literature for zero shot learning, work by Z. Akata in CVPR 2015, CVPR2016, the performance on AwA and on CUB-bird goes way above 50%, where in the current paper it is 30.57% and 8.21% at most (for the most recent survey on zero shot learning papers using attribute embeddings, please, refer to Zero-Shot Learning - The Good, the Bad and the Ugly by Xian et al, CVPR 2017). It is important to understand, why there is such a big drop in performance in one-shot learning comparing to zero-shot learning? One possible explanation is as follows: in the zero-shot learning, one has access to large training data to learn the semantic embedding (training classes). In contrary, in the proposed approach, the auto-encoder model (with 10 hidden layers) is learned using 50 training samples in AwA, and 200 images of birds (or am I missing something?). I am not sure, how can the auto-encoder model not overfit completely to the training data instances. Perhaps, one could try to explore the zero-shot learning setting, where there is a split between train and test classes: training the autoencoder model using large training dataset, and adapting the weights using single data points from test classes in one-shot learning setting.
Overall, I like the idea, so I am leaning towards accepting the paper, but the empirical evaluations are not convincing.

---

### Official Review · AnonReviewer4 · 2017-12-11
**An interesting approach to an important problem, but requires some clarifications / additional experiments in order to be convincing**

**Rating:** 5
**Confidence:** 4

**Review:**

Summary:
This paper proposes a data augmentation method for one-shot learning of image classes. This is the problem where given just one labeled image of a class, the aim is to correctly identify other images as belonging to that class as well.
The idea presented in this paper is that instead of performing data augmentation in the image space, it may be useful to perform data augmentation in a latent space whose features are more discriminative for classification. One candidate for this is the image feature space learned by a deep network. However they advocate that a better candidate is what they refer to as "semantic space" formed by embedding the (word) labels of the images according to pre-trained language models like word2vec. The reasoning here is that the image feature space may not be semantically organized so that we are not guaranteed that a small perturbation of an image vector will yield image vectors that correspond to semantically similar images (belonging to the same class). On the other hand, in this semantic space, by construction, we are guaranteed that similar concepts lie near by each other. Thus this space may constitute a better candidate for performing data augmentation by small perturbations or by nearest neighbour search around the given vector since 1) the augmented data is more likely to correspond to features of similar images as the original provided image and 2) it is more likely to thoroughly capture the intra-class variability in the augmented data.
The authors propose to first embed each image into a feature space, and then feed this learned representation into a auto-encoder that handles the projection to and from the semantic space with its encoder and decoder, respectively. Specifically, they propose to perform the augmentation on the semantic space representation, obtained from the encoder of this autoencoder. This involves producing some additional data points, either by adding noise to the projected semantic vector, or by choosing a number of that vector's nearest neighbours. The decoder then maps these new data points into feature space, obtaining in this way the image feature representations that, along with the feature representation of the original (real) image will form the batch that will be used to train the one-shot classifier.
They conduct experiments in 3 datasets where they experiment with augmentation in the image feature space by random noise, as well as the two aforementioned types of augmentation in the semantic space. They claim that these augmentation types provide orthogonal benefits and can be combined to yield superior results.

Overall I think this paper addresses an important problem in an interesting way, but there is a number of ways in which it can be improved, detailed in the comments below.

Comments:
-- Since the authors are using a pre-trained VGG for to embed each image, I'm wondering to what extent they are actually doing one-shot learning here. In other words, the test set of a dataset that is used for evaluation might contain some classes that were also present in the training set that VGG was originally trained on. It would be useful to clarify whether this is happening. Can the VGG be instead trained from scratch in an end-to-end way in this model?

-- A number of things were unclear to me with respect to the details of the training process: the feature extractor (VGG) is pre-trained. Is this finetuned during training?  If so, is this done jointly with the training of the auto-encoder? Further, is the auto-encoder trained separately or jointly with the training of the one-shot learning classifier?

-- While the authors have convinced me that data augmentation indeed significantly improves the performance in the domains considered (based on the results in Table 1 and Figure 5a), I am not convinced that augmentation in the proposed manner leads to a greater improvement than just augmenting in the image feature domain. In particular, in Table 2, where the different types of augmentation are compared against each other, we observe similar results between augmenting only in the image feature space versus augmenting only in the semantic feature space (ie we observe that "FeatG" performs similarly as "SemG" and as "SemN"). When combining multiple types of augmentation the results are better, but I'm wondering if this is because more augmented data is used overall. Specifically, the authors say that for each image they produce 5 additional "virtual" data points, but when multiple methods are combined, does this mean 5 from each method? Or 5 overall? If it's the former, the increased performance may merely be attributed to using more data. It is important to clarify this point.

-- Comparison with existing work: There has been a lot of work recently on one-shot and few-shot learning that would be interesting to compare against. In particular, mini-ImageNet is a commonly-used benchmark for this task that this approach can be applied to for comparison with recent methods that do not use data augmentation. Some examples are:
- Model-Agnostic Meta-Learning for Fast Adaptation of Deep Networks. (Finn et al.)
- Prototypical Networks for Few-shot Learning (Snell et al.)
- Matching Networks for One-shot Learning (Vinyals et al.)
- Few-Shot Learning Through an Information Retrieval Lens (Triantafillou et al.)

-- A suggestion: As future work I would be very interested to see if this method can be incorporated into common few-shot learning models to on-the-fly generate additional training examples from the "support set" of each episode that these approaches use for training.

---

### Decision · Program_Chairs · 2018-01-29
**ICLR 2018 Conference Acceptance Decision**

**Decision:**

Reject

**Comment:**

Two reviewers recommended rejection, and one was on the edge. There was no rebuttal to address the concerns and questions posed by the reviewers.